# A novel panel of *Drosophila TAFAZZIN* mutants in distinct genetic backgrounds as a resource for therapeutic testing

Kristin Richardson[ID], Robert Wessells[ID]*

Department of Physiology, Wayne State University School of Medicine, Detroit, MI, United States of America

* rwessell@med.wayne.edu

**Data Availability Statement:** All relevant data are within the paper and its Supporting Information files.

**Funding:** All funding sources are through the NIH, www.NIH.gov RO1 – R01AG059683 to RW R21 –

## Abstract

Barth Syndrome is a rare, X-linked disorder caused by mutation of the gene *TAFAZZIN* (*TAZ*). The corresponding Tafazzin protein is involved in the remodeling of cardiolipin, a phospholipid with critical roles in mitochondrial function. While recent clinical trials have been promising, there is still no cure for Barth Syndrome. Because *TAZ* is highly conserved, multiple animal and cell culture models exist for pre-clinical testing of therapeutics. However, since the same mutation in different patients can lead to different symptoms and responses to treatment, isogenized experimental models can't fully account for human disease conditions. On the other hand, isogenized animal models allow for sufficient numbers to thoroughly establish efficacy for a given genetic background. Therefore, a combined method for testing treatments in a panel of isogenized cohorts that are genetically distinct from each other would be transformative for testing emerging pre-clinical therapies. To aid in this effort, we've created a novel panel of 10 *Drosophila* lines, each with the same *TAZ* mutation in highly diverse genetic backgrounds, to serve as a helpful resource to represent natural variation in background genetics in pre-clinical studies. As a proof of principle, we test our panel here using nicotinamide riboside (NR), a treatment with established therapeutic value, to evaluate how robust this therapy is across the 10 genetic backgrounds in this novel reference panel. We find substantial variation in the response to NR across backgrounds. We expect this resource will be valuable in pre-clinical testing of emerging therapies for Barth Syndrome.

## Introduction

Barth Syndrome (BTHS) is a rare, life-threatening genetic disorder often diagnosed in infancy or early childhood depending on the severity of outward symptoms. The cardinal symptoms of BTHS can include cardiomyopathy, muscle weakness, exercise intolerance and neutropenia [1]. These cause extreme fatigue, influence the ability to eat, swallow, and maintain nutrients, and increase the risk of deadly infections like sepsis. The mutations responsible for disease occur in a gene called TAFAZZIN (*TAZ*) located on the X-chromosome [2]. Since the discovery of *TAZ* as the disease causing gene, hundreds of mutations have been identified across all

R21NS134144 to RW T32 – 2T32HL120822 to KR
*The funders had no role in study design, data
collection and analysis, decision to publish, or
preparation of the manuscript.

**Competing interests:** The authors have declared
that no competing interests exist.

exons of the gene [3, 4]. Additionally, several common splice variants have been identified, and their relative frequency compared to healthy individuals and their requirements for functional *TAZ* have been described [4–9]. *TAZ* codes for an enzyme involved in lipid metabolism; specifically in the remodeling of the mitochondrial phospholipid cardiolipin [10]. Because cardiolipin is required for stabilizing and supporting the normal functions of the electron transport chain [10, 11], defective cardiolipin remodeling impairs mitochondrial function and energy production. Since there is no cure for BTHS, medical care focuses on treating the symptoms.

As an X-linked disorder, BTHS is passed down from mother to son with a 50% likelihood that each male child will inherit the disease. It is not uncommon for more than one child in the same family to be diagnosed with BTHS. However, even siblings who inherit the exact same mutation often don't develop the same type or severity of symptoms [12]. This high phenotypic diversity between individuals is one of the major obstacles that BTHS research faces as far as developing broadly beneficial treatments. The reason for this diversity, even in siblings with similar genetics and the same inherited mutation, is likely a result of interactions with individual genetic backgrounds.

The majority of research aimed at developing treatments for BTHS has been conducted in single, isogenized, model organisms [13]. This approach doesn't account for the vast genetic diversity present in human patients which can greatly influence outcomes of treatment. Recently, a few promising therapies moved to clinical trials and are ongoing [14, 15] but initial results have been insignificant or difficult to interpret. If background genetics is a key contributing factor to differences in severity of disease and response to treatment, and these factors remain unknown, it becomes difficult to predict how they may affect the efficacy of various therapeutic options. To minimize this uncertainty and increase the chances of success during clinical trials, a research tool capable of at least partially representing that diversity in preclinical studies would be of great importance to BTHS research moving forward.

*Drosophila* has long been an ideal model organism for addressing complicated, large-scale genetic problems, including identifying genes that regulate phenotype diversity in traits such as longevity [16] and induced-exercise [17]. For these types of studies, a publicly available collection of wild-caught lines called the *Drosophila* Genetic Reference Panel (DGRP) has been isogenized and fully sequenced for the purpose of understanding the relationship between genetic variation and phenotypic outcomes [18]. Specific to BTHS, *Drosophila* mutants with deletions of the 3' region of the gene, thus lacking full-length functional *TAZ*, have been developed [8, 19]. One of these deletion mutants contains an RFP knock-in at the deletion site to facilitate crossing into other genetic backgrounds, and this was the one used in this study [20]. Each of these models has been used effectively to aid in our understanding of the progression and biology of the disease, with the goal of developing effective therapies [8, 19–21].These *Drosophila* models have been instrumental in the discovery of pharmacological targets able to rescue exercise capacity and mitochondrial deficits [20, 22]. The *TAZ* mutant line used in these studies has been extensively characterized and has several conserved phenotypes with human BTHS [21]. By introducing this mutation into selected lines from the DGRP, we have generated a panel of 10 genetically distinct *TAZ* mutants to serve as a resource for testing emerging therapeutics across highly diverse genetic backgrounds. These wild-derived lines have a high degree of genotypic and phenotypic diversity and were selected for use in our panel based on published variations in traits that are highly likely to correlate with mitochondrial function: mobility and longevity [23]. Unlike current models, this resource will allow researchers to test whether a treatment works only on some genetic backgrounds, or across a wide variety of backgrounds carrying the same mutation.

Here, we provide an initial characterization of our 10 novel DGRP-*TAZ* mutants, describing the variable phenotypic severity in endurance, mobility, and mitochondrial respiration, as

well as the first application of this panel for testing potential therapeutics. Supplementation with the NAD$^+$ precursor nicotinamide riboside (NR) was sufficient to improve both exercise and mitochondrial deficits in the original *TAZ* mutant background [20, 22]. As proof of principle for the utility of this newly developed panel as a resource for representing phenotypic diversity and variable response to treatment, we fed NR to each of the DGRP-*TAZ* lines to assess how variations in genetic background alter efficacy.

## Results

### DGRP panel baseline endurance and climbing ability is highly variable

To ensure that the large phenotypic variance in mobility previously seen in the DGRP lines [23] held up in our hands, we first assessed baseline performance via our lab-generated exercise paradigm [24, 25]. This was critical because we wanted to guarantee our panel would represent a useful range of phenotypic severity once *TAZ* mutation was introduced. We found substantial variation in both baseline endurance (Fig 1A) and climbing ability (Fig 1B). The average time to fatigue in these lines ranged from 265 to 534 minutes, averages that are within the range of variation we have previously observed in other genotypes [26]. Importantly, even the lines with the lowest endurance ran long enough to leave room for decline if *TAZ* deletion significantly impaired performance.

### Introduction of TAZ mutation reduces performance in select DGRP backgrounds

Since we already had a strong, characterized *TAZ* mutant readily available, introducing the *TAZ* deletion into our DGRP panel was conveniently done using a simple backcrossing scheme (S1 Fig). After 10 generations of backcrossing, the original background is nearly completely replaced with the new DGRP background providing 10 new, genetically distinct,

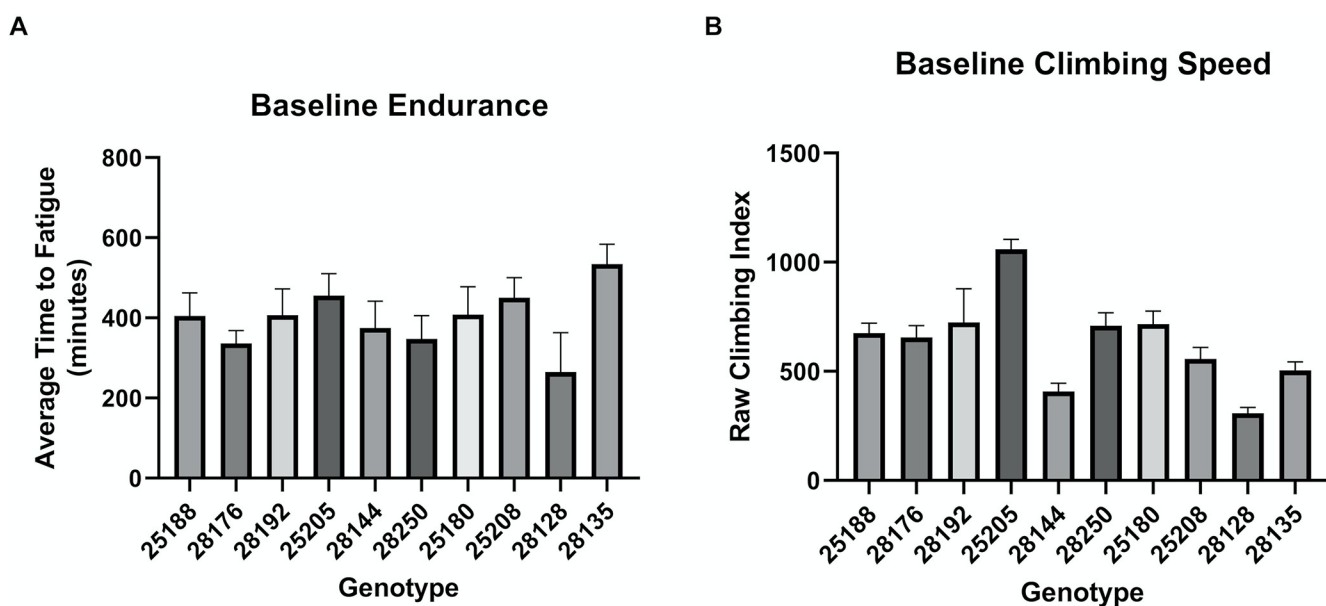

**Fig 1. Baseline performance varies considerably with genetic background.** (A) Average time to fatigue of 3-day old male flies (n = 10 vials, 200 flies) from selected genetic backgrounds, identified by Bloomington Drosophila stock center line number. Data are represented as means ± SD. (B) Climbing speed from selected genetic backgrounds, represented as raw climbing index in arbitrary units (a.u.). 3-day old flies (n = 100 flies), mean ± SD of 10 repetitions per cohort.

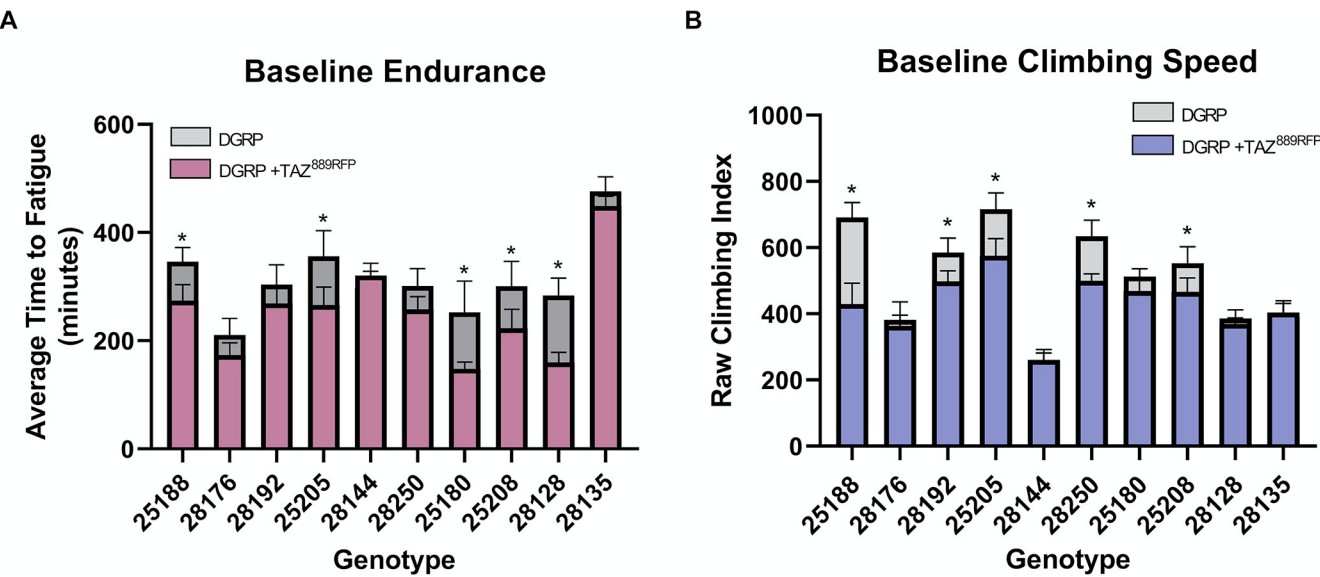

**Fig 2. Deletion of TAZ variably alters baseline performance across backgrounds.** (A) Average time to fatigue is reduced by variable amounts in selected genetic backgrounds following introduction of *TAZ* mutation. 3-day old male flies (n = 10 vials, 200 flies) means ± SD. (B) Climbing speed is reduced by *TAZ* deletion in 5 out of 10 backgrounds. 3-day old male flies (n = 100 flies), mean ± SD, statistical significance was determined using one-way ANOVA + Tukey. Only statistical comparisons between corresponding background and TAZ mutant lines are noted. *$P < 0.05$.

isogenized lines harboring the exact same *TAZ* mutation. Deletion of *TAZ* in each of the new DGRP-*TAZ* lines was verified through PCR (S1 Fig).

Introducing the *TAZ* deletion resulted in significant reductions to average endurance in 5 of the 10 DGRP-*TAZ* lines (Fig 2A). Climbing speed was reduced in 2 additional lines (5 total, 3 overlapping) following the introduction of *TAZ* mutation (Fig 2B). Interestingly, for each parameter measured there were 5 lines unaffected by the loss of *TAZ*. These findings are consistent with circumstances in humans where some individuals are affected more severely despite having similar or even identical disease-causing mutations.

## NR supplementation improves performance and mitochondrial respiration in a subset of DGRP-TAZ mutants

NR supplementation is known to benefit several disease states that have metabolic components [27] without harmful side effects [28–30]. In flies, NR improves endurance and mitochondrial deficits, specifically in the *TAZ* mutant used to generate our DGRP-*TAZ* panel [20, 22]. Because NR supplementation was sufficient to restore health in this single inbred line, we wanted to use it as a treatment to demonstrate the utility of our DGRP-*TAZ* panel for testing promising therapeutic options on genetically diverse subjects.

While pre-treatment assessments of endurance and climbing ability were done on 3-day old flies (Fig 1A and 1B), established NR supplementation methods require 10 days of feeding prior to assessing the effect on performance. Therefore, NR supplementation data is taken from 10-day old flies fed NR throughout adult life (S2 Fig). Because Barth Syndrome is progressive, the effects of *TAZ* deletion further impacted performance by day 10, resulting in a higher number of affected lines on day 10 than on day 3. Compared to controls at day 10, there were 7 DGRP-*TAZ* lines with significantly reduced endurance, and 9 DGRP-*TAZ* lines with reduced climbing speed (Table 1). Of these, 6 were accompanied by impaired mitochondrial respiration, as measured by respiratory control ratio (RCR = ratio of ADP stimulated

**Table 1. Summary of day 10 endurance, speed and mitochondrial respiration data with NR feeding.**

| Genotype | Day 10 Table 1 | | | | | |
|---|---|---|---|---|---|---|
| | Endurance Phenotype | Climbing Phenotype | Mitochondrial RCR Phenotype | NR Improved: Endurance | NR Improved: Climbing | NR Improved: Mitochondrial RCR |
| 25188 | Y | Y | Y | Y | Y | Y |
| 28176 | Y | Y | Y | Y | Y | Y |
| 28192 | Y | Y | N | N | N | N |
| 25205 | N | Y | Y | N | Y | N |
| 28144 | N | Y | Y | N | N | N |
| 28250 | Y | N | N | N | N | N |
| 25180 | Y | Y | N | N | N | N |
| 25208 | Y | Y | N | N | Y | Y |
| 28128 | Y | Y | Y | N | N | N |
| 28135 | N | Y | Y | N | N | Y |

Summary of significant phenotypic changes in exercise performance and mitochondrial respiration after *TAZ* deletion across background lines. Lines where NR supplementation caused a significant phenotypic rescue are marked with a Y. Those where NR failed to rescue are marked with an N. Complete data appears in S2 Fig.

respiration rate:ADP depleted respiration rate). The degree to which each parameter was reduced with loss of *TAZ* as a percent change was highly variable, ranging from mild at 12% to severe at 53% (Table 2). NR supplementation improved endurance in 2 lines, climbing speed in 4 lines, and mitochondrial function in 4 lines (Table 1). The percentage of the deficit that was rescued by NR supplementation ranged from 17% to 100% (Table 2). These differences in percent rescue by NR supplementation were not due to increased intake of NR, as feeding rates within lines benefited by NR supplementation on any of the three parameters measured were not different (S3 Fig).

## Performance correlates with mitochondrial function

The results across the 10 DGRP-*TAZ* lines are clearly complex, with some lines gaining severe impairments to performance even without impairments to mitochondrial function and vice

**Table 2. Summary of % change in endurance, climbing and mitochondrial respiration data with NR feeding.**

| Genotype | Day 10 Table 2 | | | | | |
|---|---|---|---|---|---|---|
| | Endurance % Change | Climbing % Change | Mitochondrial % Change | NR Improved: Endurance Deficit | NR Improved: Climbing Deficit | NR Improved: Mitochondrial Deficit |
| 25188 | - | -40.35% | -44.29% | 75.81% | 17.78% | 63.66% |
| 28176 | -52.97% | -43.90% | -51.58% | 74.41% | 20.15% | 120.88% |
| 28192 | -30.22% | -27.22% | - | - | - | - |
| 25205 | - | -35.64% | -34.09% | - | 41.92% | - |
| 28144 | - | -15.22% | -42.59% | - | - | - |
| 28250 | -23.46% | - | - | - | - | - |
| 25180 | -42.80% | -12.05% | - | - | - | - |
| 25208 | -43.15% | -42.19% | - | - | 35.67% | - |
| 28128 | -53.88% | -29.49% | -46.90% | - | - | - |
| 28135 | - | -28.45% | -31.04% | - | - | 109.67% |

Table 2 quantifies reductions in performance as % change and rescue as a % of the total deficit. The %change is calculated as (average value in mutant/average value in WT)/(average value in WT)*100, and % of deficit rescued for each parameter is calculated as (%change of mutant—%change of mutant +NR)/(%change of mutant)*100.

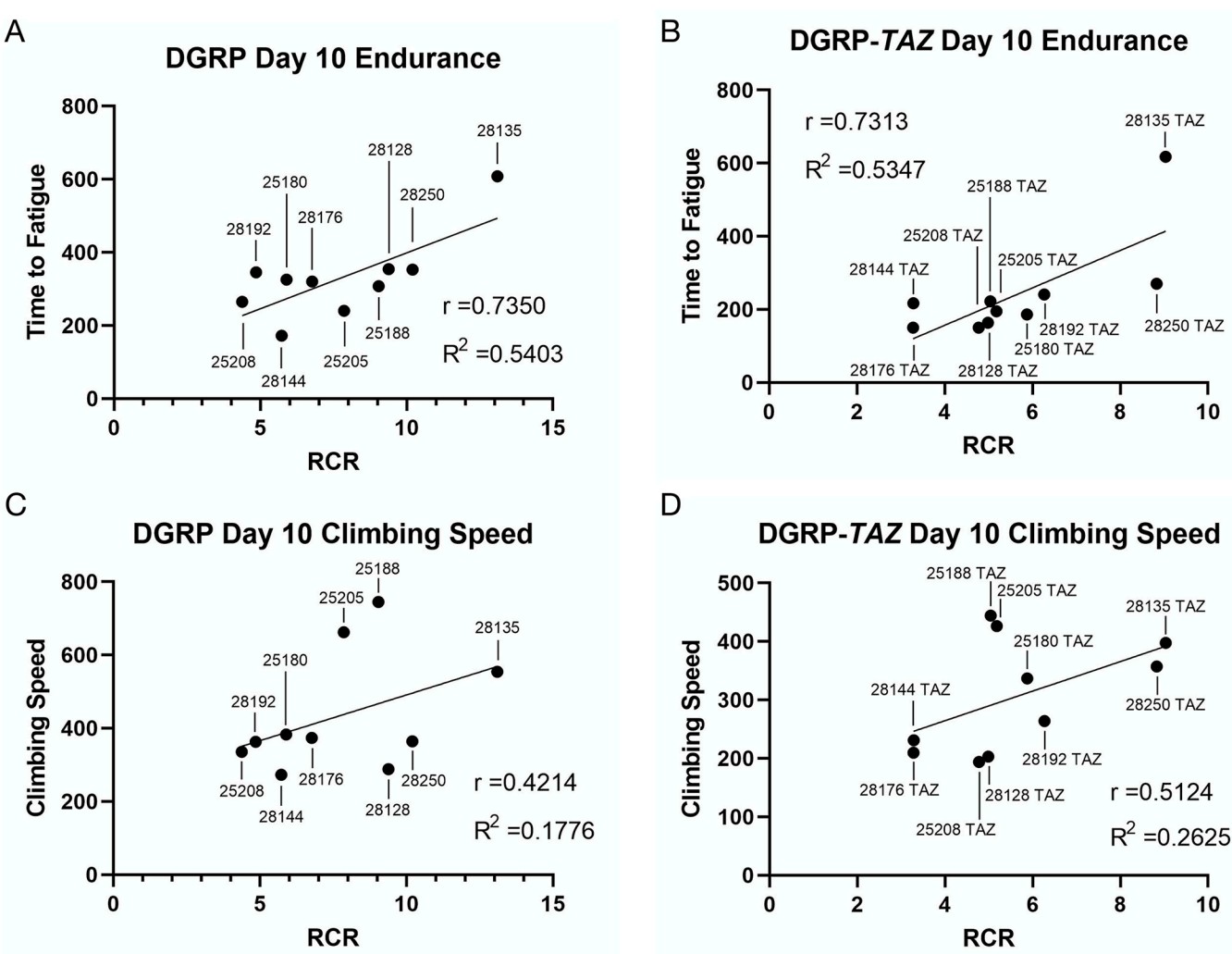

**Fig 3. Mitochondrial RCR correlates with endurance but not speed.** Correlation between endurance and RCR in (A) background lines and (B) *TAZ* mutants in the same backgrounds. There is a strong, positive linear relationship between endurance and RCR in both cohorts (r>0.7). Each point represents average data from a single line. DGRP endurance vs RCR was significant by Pearson's Correlation $P = 0.0154$, DGRP-*TAZ* endurance vs RCR $P = 0.0163$. (C) Correlation between climbing speed and RCR in background controls and (D) *TAZ* mutants in the same backgrounds. There is a moderate, positive linear relationship between climbing speed and RCR in background controls (r>0.5), and a low, positive linear correlation in mutants (r>0.03). Neither relationship reaches statistical significance by Pearson's Correlation ($P = 0.2251$ and $P = 0.1300$ respectively).

versa (Tables 1 and 2). To help make sense of the link between mobility outputs and mitochondria, we asked whether endurance or climbing ability were correlated with mitochondrial RCR. We hypothesized that due to the sustained energy demand of endurance, a stronger correlation with RCR would exist in endurance than in climbing speed. For both the DGRP background lines and the DGRP-*TAZ* lines, endurance had strong positive correlation with RCR (Fig 3A and 3B). In both cohorts, a higher RCR predicted a longer average time to fatigue. Climbing speed trended toward a weak positive correlation with RCR in both cohorts (Fig 3C and 3D), although neither p-value was significant. The lack of a strong climbing speed phenotype here is in line with findings in the original *TAZ* mutant in a w[1118] background [21]).

## Discussion

In building this novel panel of DGRP-*TAZ* lines, we have shown that 1) deletion of *TAZ* negatively impacts certain backgrounds more than others, and 2) treatment efficacy can be altered depending on genetic background. The magnitude or existence of rescue following NR supplementation does not appear to be reliably linked to severity of deficit, as some lines with very severe impairment did not benefit from NR supplementation while others with similar deficits did (Table 2 and S2 Fig). Additionally, while there was some overlap between lines that gained an endurance or climbing phenotype following the introduction of mutant *TAZ*, they are not identical; perhaps highlighting a potential difference in the energetic requirements for each activity (i.e., sustained vs brief). It may be that because climbing speed assays are short term rather than prolonged periods of exertion, flies with less efficient mitochondria may still be able to function to a degree that would appear as normal during this short time.

Because BTHS is primarily a disease of the mitochondria, it's interesting that performance deficits existed in a few lines without a reduction in mitochondrial RCR (Table 2). Perhaps in these backgrounds there is a compensatory mechanism able to boost the mitochondria, but unable to completely eliminate the physiological impacts of the disease. However, with only 10 lines, addressing what those genetic modifiers may be is outside the scope of this study. Additionally, in one of the lines, supplementation with NR was able to rescue 100% of the mitochondrial RCR phenotype without improving physiological function in either endurance or mobility. In this case, it's possible the mitochondrial improvements weren't enough to overcome physiological demand. All lines that had a significant mitochondrial RCR phenotype also had either an endurance or a climbing speed reduction (Table 2). In each of these scenarios, it's important to consider that measurements such as RCR are "snap-shots" of mitochondrial respiration in isolated mitochondria with an abundance of substrate availability [31]). These conditions don't necessarily fully represent mitochondrial efficiency in the whole animal and therefore require caution when making causal interpretations [31].

Of the three parameters measured, NR supplementation provided the largest percent benefit to endurance and mitochondrial RCR, and only mild improvements to climbing speed (Table 2). Furthermore, NR supplementation was only able to rescue deficits within all three parameters in 2 of the 10 DGRP-TAZ lines. These 2 lines had relatively average reductions in performance and mitochondrial RCR following *TAZ* deletion, so it is not obvious why they responded so well to NR treatment aside from the undefined influences of their genetic background. In the original *TAZ* mutant, NR supplementation was able to rescue endurance, climbing speed and mitochondrial RCR without any significant impact on WT background controls [20]. Here, for the most part, NR had no effect on WT performance or was mildly beneficial (S2 Fig). However, in some lines, NR was actually harmful to both background controls and *TAZ* mutants (S2 Fig). These negative effects could not have been revealed using only one single isogenized line, further underlining the need for experiments using resources such as the new panel presented here moving forward.

BTHS is not alone in the need for applications such as this. Background-dependent effects can be seen in a variety of diseases and therapeutic efficacies [26, 32–35]). For example, in mouse models of neuromuscular disorders such as Amyotrophic Lateral Sclerosis and Spinal Muscular Atrophy, backcrossing known disease-causing mutations into different strains alters survival and disease severity [35, 36]. In these studies, physical impairments (strength, weight loss, loss of motor neurons and NMJ pathology) begin at an earlier age and lifespans are shortened in certain strains compared to others, in some cases even with no significant differences in the amount of disease-associated protein [35]. In regard to background-dependent treatment effects, the efficacy of deep brain stimulation for the treatment of Parkinson's Disease,

which can be highly effective [37, 38], can also be highly variable between patients [34]. The same phenomena can be seen in diabetes [33] and kidney disease [39]. Despite the high prevalence of background-dependent phenotypes and the large number of GWAS studies aimed at identifying secondary loci that influence disease [40, 41], genetic diversity is still rarely accounted for in preclinical studies. Of particular benefit from these considerations may be rare diseases with a limited patient population available for GWAS studies or clinical testing. Pre-clinical resources using genetically diverse model organisms such as described in this paper could serve as a useful pre-screening for efficacy of potential therapies across backgrounds in these scenarios.

Based on the data presented in this study, there are clear background-dependent effects on phenotype severity in *Drosophila* models of BTHS, as there are in human BTHS patients. Each of these DGRP-*TAZ* lines carries the exact same mutation, however, the degree to which each was impaired, as well as the degree to which treatment with NR was able to rescue those impairments, differed greatly. Although our panel is not large enough to dissect out the genetic modifiers of this phenotypic diversity, and is instead intended to represent that diversity in pre-clinical studies, these data highlight the need for extending treatment-focused studies across multiple genetic backgrounds to assess potential variation in treatment response. Adopting practices that help account for individual variations across a population would enable greater predictability of clinical outcomes as we work toward making personalized medicine routine.

## Experimental procedures

**Generation of DGRP panel.** Each of the newly generated DGRP-*TAZ* lines were created by backcrossing an existing *TAZ* mutant line in a w[1118] background with an RFP marker knocked in to the deleted region [20, 21] (*TAZ*[889], WellGenetics) to *Drosophila* Genome Reference Panel lines selected based on their variable running behaviors. Because crossing over doesn't occur in male flies, females were used to carry the mutation through 10 generations of crosses. Flies containing the deletion in each generation were easily selected as heterozygotes using the visible RFP marker. Due to homozygous male infertility in the w[1118]-*TAZ* mutant, second chromosome balancers were introduced to each line, without affecting other chromosomes. Homozygous escapers were readily collectible from each background for experimental assessments.

**qRT-PCR analysis.** Knockdown of *TAFAZZIN* expression for each of the newly generated *TAZ* mutant lines was confirmed through qRT-PCR analysis. RNA was isolated from 20 whole flies using TRIZOL (Invitrogen), and plates were set up using Power SYBR Green Master Mix (Applied Biosystems, Waltham, MA, USA) and ran on a StudioQuant 3 Real-time PCR System (Thermo Fisher Scientific). Each well contained the following: template RNA (4μl of 25ng/μl concentration), forward and reverse primers (2μl at 1μM concentration), Power SYBR Green PCR Master Mix (10μl), reverse transcriptase (0.1μl), RNase inhibitor (0.025μl) and DNase free dH$_2$O (3.875μl). For each genotype, 2–3 independent biological replicates were examined and divided into 3 technical replicates. Resulting mRNA data were normalized to *Act5C*. Primers used were as follows:

*Act5C* F, 5′–CGCAGAGCAAGCGTGGTA–3′
*Act5C* R, 5′–GTGCCACACGCAGCTCAT–3′
*Tafazzin* F, 5′–GATTCTCAGGTGCCCAATG–3′
*Tafazzin* R, 5′–TCGTCAGAGATCTATGGCG–3′.

**Climbing speed.** Climbing speed was assessed by rapid iterative negative geotaxis (RING) [42]. 5 vials containing 20 flies each (n = 100) were set up in a RING apparatus and tapped

down to the bottom of their vials. After 2 seconds a digital image was taken to capture the height of the flies in their vials. 10 images were taken per cohort at day 3 for baseline measurements or day 10 in NR fed experiments. Images were analyzed with ImageJ and averages were calculated and graphed using GraphPad prism software. Raw (non-normalized) climbing index is defined and calculated as follows: in ImageJ, a height value in pixel units (arbitrary units) is assigned for each individual fly (n = 100) in a single image. The height value provided for all 100 flies in an image is averaged to generate a single value for each of the 10 images taken at the assessed timepoints. Those 10 average values are graphed to make up the raw climbing index, or non-normalized average climbing height in arbitrary units.

**Baseline endurance.** Baseline endurance was assessed using the PowerTower *Drosophila* exercise platform [25]. Endurance was graphed as average time to fatigue across 10 vials containing 20 flies each. A vial was considered "fatigued" when fewer than 20% of flies were still climbing more than 1 centimeter up the side of their vial [42]. Endurance was assessed at baseline (day 3) and day 10 timepoints.

**NR feeding.** 60uL of 1mM NR (dissolved in ddH$_2$O) was pipetted directly onto vials and allowed to dry before flipping flies onto it. This concentration was chosen based on success in previously published work (20). Flies were maintained on NR or vehicle (ddH20) supplemented food for 10 days prior to completing functional assays. Fresh vials were provided every 2 days. NR (TRU NIAGEN) was obtained from ChromaDex.

**Feeding rate.** Feeding rate was measured by observing 8 vials of 10 flies each for a total of 3 seconds, 10 times and recording the number of flies within each vial that were feeding, as known by proboscis extension and associated "bobbing" as previously described [43, 44]. The average number of flies feeding in each of the 10 observations was calculated and graphed as percent feeding.

**Mitochondrial respiration analysis.** Mitochondrial respiration was assessed by calculating the respiratory control ratio between state III (ADP stimulated) and State IV (ADP depleted) respiration rates. Mitochondria were isolated from whole fly lysates (60 flies per replicate) by centrifugation and protein content was measured by BCA. A Clarke electrode was used to measure the respiration rates of isolated mitochondria within 2 hours of isolation. 1,200mg ±150mg (10uL of isolated sample) was loaded into the calibrated electrode and measurements for corresponding cohorts were all done on the same day with the same calibration and alternating between DGRP or DGRP-*TAZ* groups to prevent conclusions from being affected by wait times. Average RCR was calculated from 3 biological replicates per cohort.

## Supporting information

**S1 Fig. Backcrossing scheme and validation of knockdown by PCR.** (A) Schematic representation of backcrossing scheme used to generate the 10 new DGRP-TAZ mutant lines. Created with BioRender.com (B) PCR results confirming reduced TAZ expression in each of the 10 backcrossed DGRP background lines, two tailed, unpaired Student's t-test. *P<0.05, **P<0.01, ***P<0.001, ****P<0.0001.
(TIF)

**S2 Fig. NR supplementation improves performance and mitochondrial respiration in some DGRP-TAZ mutants.** Day 10 average endurance (red, n = 10 vials, 200 flies), climbing speed (blue, n = 100 flies) and mitochondrial RCR (green, n = 3 biological replicates) with NR supplementation. Two-way ANOVA + Tukey. *P<0.05, **P<0.01, ***P<0.001, ****P<0.0001. Despite the large separation of average RCR, the benefit provided to mitochondrial RCR by NR supplementation related to line 25208 showed a large degree of variation across repetitions in 2 of the 4 cohorts (25208 +NR and 25208 *TAZ*), therefore resulting in

modest p-values or non-significant results: 25208 vs 25208 +NR: *P = 0.033, 25208 vs 25208 *TAZ*: P = 0.99, 25208 vs 25208 *TAZ* +NR: *P = 0.046, 25208 +NR vs 25208 TAZ: *P = 0.044, 25208 +NR vs 25208 *TAZ* +NR: P = 0.98, 25208 *TAZ* vs 25208 *TAZ* +NR: P = 0.062.
(ZIP)

**S3 Fig. Benefits provided by NR supplementation are not due to increase in NR intake.** Lines included in this figure gained improvements in either endurance, climbing speed or mitochondrial RCR after 10 days of NR feeding. No significant difference was seen in feeding rates between these lines. (n = 10 vials, 100 flies) One-Way ANOVA, P = 0.7186.
(TIF)

## Author Contributions

**Conceptualization:** Kristin Richardson, Robert Wessells.

**Data curation:** Kristin Richardson, Robert Wessells.

**Formal analysis:** Kristin Richardson, Robert Wessells.

**Funding acquisition:** Robert Wessells.

**Methodology:** Robert Wessells.

**Project administration:** Robert Wessells.

**Resources:** Robert Wessells.

**Validation:** Kristin Richardson, Robert Wessells.

**Visualization:** Kristin Richardson, Robert Wessells.

**Writing – original draft:** Kristin Richardson.

**Writing – review & editing:** Kristin Richardson, Robert Wessells.

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
