## [Decision Letter · Decision Letter 0]

9 May 2023

PONE-D-23-10039

A novel panel of Drosophila TAFAZZIN mutants in distinct genetic backgrounds as a resource for therapeutic testing

PLOS ONE

Dear Dr. Wessells,

Thank you for submitting your manuscript to PLOS ONE. After careful consideration, we feel that it has merit but does not fully meet PLOS ONE’s publication criteria as it currently stands. Therefore, we invite you to submit a revised version of the manuscript that addresses the points raised during the review process.

We look forward to receiving your revised manuscript.

Kind regards,

Giovanni Messina

Academic Editor

PLOS ONE

Journal Requirements:

Additional Editor Comments:

Dear Authors,

I have ticked 'Major Revision' but actually it is really hard to me to take a decision based on these two opposite opinions about your paper. Since that the Reviewer 1 is really harsh without going into the details of its comments, I would like to give you the opportunity to reply to it.

In the meanwhile, you can address the point of Reviewer 2 so I can take a final decision later.

Thank you.

Best regards

Giovanni Messina

Reviewers' comments:

Reviewer's Responses to Questions

**Comments to the Author**

1. Is the manuscript technically sound, and do the data support the conclusions?

Reviewer #1: No

Reviewer #2: Yes

2. Has the statistical analysis been performed appropriately and rigorously? 

Reviewer #1: Yes

Reviewer #2: Yes

3. Have the authors made all data underlying the findings in their manuscript fully available?

Reviewer #1: Yes

Reviewer #2: Yes

4. Is the manuscript presented in an intelligible fashion and written in standard English?

Reviewer #1: Yes

Reviewer #2: Yes

5. Review Comments to the Author

Reviewer #1: This is a poorly designed study with no clearly defined goals and, consequently, no meaningful conclusion. The authors began by pointing out that genetic background influences both the severity of diseases (e.g. Barth syndrome) and the efficacy of treatments. They then put TAZ mutation into 10 different fly strains of defined genetic background and tested NR treatment on them. Because the study is so underpowered, they could not reach any conclusion beyond their starting point.

Reviewer #2: In this manuscript, Richardson and Wessells investigate how genetic background impact phenotypes caused by deletion of Tafazzin, the Barth Syndrome gene, in 10 lines from the Drosophila melanogaster Genetic Reference Panel, and their response to treatment with nicotinamide riboside (NR), which the authors previously shown beneficial in a fly model of Barth Syndrome. The authors assessed exercise and climbing performance, and also RCR (ADP stimulated:depleted respiration rate ratio) as an estimate of mitochondrial respiration, and found that, in the different fly lines i) the strength of the phenotypes as variable; ii) RCR correlated with exercise performance iii) the strength of response to NR was also variable.

The study provides an interesting strategy to assess the impact of genetic diversity in preclinical studies. I have few concerns that have to be addressed before publication.

- Introduction: please provide more details about the kind of Tafazzin mutations in Barth Syndrome and available fly models.

- Lines 119-120 and 179-181: please provide data or reference of publication demonstrating that fly endurance and climbing involve different energetic requirements (i.e., sustained vs brief), otherwise remove these sentences or move them to the discussion section.

- Lines 177-179: please correct references to Fig.3 panels.

- Fig.3: please annotate fly lines in the dot plot.

- In the Experimental procedures section, please detail how the “raw climbing index” was calculated.

- In the Experimental procedures section, please include detailed description of PCR assay and sequences of the primers used in Fig.S1.

- Fig.S2: please provide statistics for line 25208 mitochondrial respiration rescue by NR.

6. PLOS authors have the option to publish the peer review history of their article (what does this mean?). If published, this will include your full peer review and any attached files.

Reviewer #1: No

Reviewer #2: No

---

## [Author Response · Author response to Decision Letter 0]

15 May 2023

We thank the reviewers for their comments, as they have greatly improved the manuscript. Detailed descriptions of changes are outlined below:

Reviewer 1:

We agree with the reviewer that the number of lines in this study is not sufficient to perform GWAS analysis or identify specific modifier loci. However, this was not our purpose in the study. Rather, our goal was to generate a novel resource to facilitate rapid testing of potential interventions in a wide range of genetic backgrounds. In this way, potential therapies can be vetted early in the process through inexpensive fly trials before human testing. Pre-testing in these carefully generated and curated fly lines will allow potential therapies to be sorted into categories based on whether they are widely effective or are more effective in particular backgrounds. We feel this resource will be a significant help to the Barth research community in the future, whereas other strategies will need to be employed to identify specific modifier loci. The generation and testing of such a novel resource seems to us to be well within the mission of the PLoS ONE journal, and as such, we feel strongly that the manuscript should be considered for publication therein. 

We have edited the text to clarify and emphasize the use of this study as a validation of a novel resource for BTHS researchers, and not as a means to actually identify modifiers of disease. (Lines: 93-95, 217-218, 239, 255-256, 263-264).

Reviewer 2:

• Introduction: please provide more details about the kind of Tafazzin mutations in Barth Syndrome and available fly models.

Please see Lines 39-43 and 72-77: Text outlining current Drosophila models of barth syndrome as well as literature on known mutations and their characterizations has been added.

• Lines 119-120 and 179-181: please provide data or reference of publication demonstrating that fly endurance and climbing involve different energetic requirements (i.e., sustained vs brief), otherwise remove these sentences or move them to the discussion section.

We have moved the referred to text into the discussion as suggested (Lines 206-212). We also clarify in our Methods section that speed is measured across 2 seconds (line 300), whereas endurance is measured until exhaustion (line 312), which typically requires hundreds of minutes (see Figures 1, 2 and supplemental).

• Lines 177-179: please correct references to Fig.3 panels.

Call outs to Figure 3 (now Lines 185 and 187) have been edited to match Figure 3 legend, and panel arrangement has been modified to correspond with those changes.

• Fig.3: please annotate fly lines in the dot plot.

Each data point has been labeled with the corresponding fly line in the revised Figure 3. We thank the reviewer for this suggestion, as it improves the figure substantially.

• In the Experimental procedures section, please detail how the “raw climbing index” was calculated.

Lines 303-308: Text has been added in Methods to describe the calculation of raw climbing index for a given genotype. Generally, the term raw we use to mean the data for each group is not normalized to its starting point. In cases where data is normalized, we include the word “normalized” in the label.

• In the Experimental procedures section, please include detailed description of PCR assay and sequences of the primers used in Fig.S1.

We thank the reviewer for catching this oversight. Description now appears in lines 281-295 in experimental procedures section.

• Fig.S2: please provide statistics for line 25208 mitochondrial respiration rescue by NR.

The large variation in RCR values across repetitions is the reason for the visibly large separation of cohorts being associated with a modest p-value for this line. Along with the statistical tests completed, the exact p-value for this line and a short description have been added to the figure legend for Figure S2 (Lines 466-471).

---

## [Editor Report · Decision Letter 1]

16 May 2023

A novel panel of Drosophila TAFAZZIN mutants in distinct genetic backgrounds as a resource for therapeutic testing

PONE-D-23-10039R1

Dear Dr. Robert Wessells,

We’re pleased to inform you that your manuscript has been judged scientifically suitable for publication and will be formally accepted for publication once it meets all outstanding technical requirements.

Kind regards,

Giovanni Messina

Academic Editor

PLOS ONE

---

## [Editor Report · Acceptance letter]

19 May 2023

PONE-D-23-10039R1 

A novel panel of *Drosophila TAFAZZIN* mutants in distinct genetic backgrounds as a resource for therapeutic testing 

Dear Dr. Wessells:

I'm pleased to inform you that your manuscript has been deemed suitable for publication in PLOS ONE. Congratulations! Your manuscript is now with our production department. 

Kind regards, 

on behalf of

Dr. Giovanni Messina 

Academic Editor

PLOS ONE